# Smooth doubly curved origami shells with reprogrammable rigidity

Morad Mirzajanzadeh ⬡ & Damiano Pasini ⬡ ✉

Origami tessellations can transform flat sheets into curved yet inherently compliant surfaces that only approximate curvature and are unable to reconcile a fundamental trade-off among load-bearing capacity, curvature precision, and stiffness reprogrammability. We resolve this conflict by introducing a tileable crease pattern that folds into smooth, doubly curved shapes, enabling structural locking with minimal sagging under load. Solving an inverse problem, we compute fold patterns that match prescribed smooth surfaces with double, variable, and constant curvature. By strategically embedding tendons with varying pre-tension, we demonstrate reversible transformations from ultrasoft, formless states into rigid, load-bearing structures with in-situ tunable stiffness spanning orders of magnitude. This work unlocks a paradigm for folding doubly curved origami metamaterials, enabling flat-pack transport and scalable deployment of smooth, load-bearing shells.

Rigid, smoothly curved surfaces are essential across science and engineering, impacting fields from aerospace and product design to optics[1,2] and biomechanics[3–5]. A material that needs to be dynamically shaped into a smoothly curved surface often requires reconciling antagonistic functions, such as morphing its shape and/or adapting its properties without compromising rigidity and smoothness in the deployed state. Typically, the trade-off between rigidity, curvature smoothness, and property tunability cannot be attained in current technology, including wearable supports and exoskeletons[6–9], deployable orthopedic implants[10], soft robotics[11,12], and deployable solar sails[13] or antennas[14]. Soft robotic metamaterials, for instance, often rely on either inflatable systems with soft skins[15] or origami structures featuring jagged, corrugated textures[16,17]. The former lacks surface rigidity, whereas the latter lacks surface smoothness. Soft skins are also vulnerable to abrasion in rough environments, such as those encountered in planetary exploration. Additionally, jagged surfaces are unsuitable for applications requiring low friction for drag reduction, for example, in swimming robots. Moreover, conformal curved contours are often necessary in orthopedic implants to match the complex geometries of curved bones. While recent origami-based deployable implants have enabled compact insertion and in situ expansion, their designs typically deploy into faceted forms that cannot form smooth, doubly curved geometries, a critical requirement for

conformal, load-bearing integration of implants in complex anatomical sites[10].

Similar challenges arise in wearable assistive technology and exoskeletons, which must conform to the curved shape of the human body while providing structural support. Existing concepts, on the other hand, often rely on discrete folds[7] or rigid, chainmail-like elements[6,9], resulting in non-smooth surfaces that can cause discomfort or even pain during prolonged wear or when compressed against the body. Currently, a technology that can deploy into a shell tessellation with a smoothly curved surface and tunable load-bearing capacity remains an unresolved challenge across various domains.

Origami principles offer a promising solution for creating deployable shells with desired curvature[18–28] and metamaterials with programmable stiffness[29–33] at multiple length scales. However, deployable origami shells composed of thin sheets are often floppy under mechanical loads due to the existence of zero-energy modes and the additional compliance of their non-rigid sheets; thin sheets can bend and twist easily, especially in origami incorporating curved creases or deformable panels. On the other hand, existing tessellable concepts typically rely on straight-crease patterns to deploy into a piecewise-linear approximation of curvature, with precision dependent on the pattern size[18–28]. Although refining the crease pattern size improves the smoothness of the approximated curves, it inevitably decreases the global thickness of the origami shell. Since flexural

---

Department of Mechanical Engineering, McGill University, Montreal, QC, Canada. ✉e-mail: damiano.pasini@mcgill.ca

rigidity scales cubically with thickness, this reduction significantly compromises the load-bearing capacity of the shell, thereby creating an inherent trade-off between curvature smoothness and load-bearing capacity.

Curved creases, by contrast, can naturally generate folded structures with smoothly curved surfaces. Primarily explored in art[34–38]—beginning with the Bauhaus experiments of Josef Albers and the systematic explorations of David Huffman, and later expanded through the swirling pleated sculptures of Erik and Martin Demaine—curved-crease forms have more recently found practical applications in engineering[39–42]. In a previous work[42], the authors introduced a hybrid curved-straight crease pattern, geometrically constructed to enable tiling into cylindrical surfaces with a prescribed global curvature of constant value. The intrinsic nature of this pattern and its kinematic constraints, however, make it impossible to generate smooth doubly-curved surfaces, fold into shapes with varying curvature, and deploy structural shells with reprogrammable flexural rigidity. These properties have significant implications that are crucial in a broad range of applications, such as doubly curved antenna reflectors for space, tunable and deployable curved structures, ergonomically shaped exoskeletons and wearables, aerodynamically shaped bodies for automotive and aerospace, and reconfigurable doubly curved soft robots with adaptive variable curvature.

One strategy for generating stiffness in deployable structures is to systematically integrate a network of pre-stretched tendons that partially or fully constrain the folding degrees of freedom, akin to cables in a structural tensegrity system[43–45]. Cables have previously been employed to control folded geometry[46] or induce mechanical bistability[47] in origami structures. More recently, tensegrity principles have been demonstrated effective for enhancing the axial stiffness during the deployment of single-degree-of-freedom, collapsible, straight-crease origami structures, such as Miura, Kresling, and modular Miura tubes[48,49]. Most of them feature unidirectionally tiled, rigid-foldable origami patterns, which primarily resist compression and tension only. These concepts, however, cannot be easily extended to in-plane origami tessellations made of bendable panels, which can fold into shell structures capable of resisting bending actions and multi-axial forces, a common scenario encountered in everyday technology.

In this work, we introduce the doubly curved lens-box, a lockable and rigid-ruling foldable[50] hybrid crease pattern that combines curved and straight creases. Once deployed in its locked state, its tessellated pattern conforms to a doubly curved surface which is smooth in one direction and piecewise linear in the other. Unlike existing concepts that are floppy, our doubly curved lens-box is supplemented with an array of tendons, here denoting cable-like elements that resist extension only. These tendons can stiffen the folded pattern by contraction, reminiscent of cables in a tensegrity structure[43–49]. Since our origami building block tessellates into a smooth, continuous surface, the resulting origami shell forms a periodic metamaterial whose effective properties can be tailored via geometric design and actively reprogrammed through tendon pre-tension post fabrication. The tendons guide folding and enable the underlying folded pattern to shift reversibly and continuously from a relaxed configuration to a rigid, multiaxially load-bearing state. This evokes phase transformation in solids, where the amount of tendon stretching, as opposed to the transition temperature, governs folding and property tunability, a process that squeezes adjacent building blocks in a manner analogous to the jamming transition[51]. The concept introduced in this work, on the other hand, does not rely on gravity[51], fluid pressure[9], and environmental stimuli.

## Results
### Geometry of the *doubly curved lens-box* pattern
Our doubly curved lens-box origami pattern comprises two types of distinct yet complementary building blocks (Fig. 1a): curved-crease lens units (greenish yellow shades), which resemble a lens shape, and flat-foldable straight-crease waterbomb connector units (blue shades), which join the straight creases of adjacent lens units. Each lens unit consists of two reflected, non-symmetric curved arcs enclosing a middle lens panel (M), separating an upper leg panel (U) from a lower leg panel (L). Hybrid-crease unit cells, formed by merging lens-units and two dissimilar-size waterbomb connectors, can tile the entire plane. Upon folding, their shared linear edges form straight creases, whereas the two-dimensional arc creases (Fig. 1a-top) deploy into their three-dimensional curved counterparts (Fig. 1a-bottom), generating an array of out-of-plane arched surfaces joined by flat panels. At a specific folding configuration, when the facets of each connector unit become coplanar (i.e., flat-folded), the origami reaches a locked state. The geometric interplay between connectors and lens units of varying sizes and shapes yields a doubly curved tessellated surface that can be smooth along the lens direction (blue curvature, $\kappa_I$) and a piecewise-linear approximation of uniform curvature along the orthogonal direction (orange curvature, $\kappa_{II}$). It also enables the formation of variable curvature geometries.

To understand the folding kinematics of our doubly curved pattern, we idealize its deformation. We recall first that a straight-crease pattern folds through the mere rotation of its hinges (creases) without flexing its faces, a property known as rigid-foldability. In contrast, folding a curved-crease pattern requires the additional deflection of its panels. For developable curved-crease lens units, we assume a sheet of vanishing thickness deforms only through bending. This is underpinned by the energy scaling law: the stretching energy scales with the panel thickness $t$, while the bending energy scales with $t^3$, making bending the dominant form of deformation. This assumption enables us to describe the kinematic of the lens unit by segmenting each curved panel into a non-crossing ruled surface (Fig. 1b), forming the rulings, a family of rule segments (see Supplementary Note 1). If the rulings remain unaltered during the entire folding process, the curved-crease is rigid-foldable[50]. To distinguish this from the rigid-foldability definition used in straight-crease origami, we hereafter refer to it as rigid-ruling foldability. Proving the existence of an invariant ruling pattern is essential for describing folding and achieving rigid-ruling foldability. This property has significant implications as it enables the use of even non-deformable materials to create the discretized, ruled segmentation version of the rigid-ruling foldable origami structures that transition from a flat state to a locked, approximately curved shape.

We demonstrate the existence of the ruling pattern and the folded geometry shown in Fig. 1b using theories of general curved-crease folding[50,52–54], insights into the ruling segmentation developed in our earlier work on the hybrid Miura lens-box crease pattern[42], and differential geometry under specific assumptions (see Supplementary Notes 1 and 2). We assume folding mapping to be isometric, i.e., a length-preserving transformation, with no twisting deformation in the lens panel M. The rulings within the lens panel M are thus parallel segments, and the geometry of the folded lens unit remains symmetric with respect to the plane bisecting M. Additionally, we assume that curved creases fold smoothly, generating surfaces with $C^2$ continuity (second derivative) and kink-free folded creases. This assumption is justified by the energetic cost of non-smooth folding, which would require the formation of kinks or additional creases. These configurations are less energetically favorable because the bending energy required for smooth folding is much lower than the stretching energy needed to create kinks or new creases. Our experimental observations qualitatively support this assumption.

The folded geometry of the generalized lens unit is constructed by establishing the relations between pairs of points on creases connected through rule segments (see Supplementary Note 2.1). The unfolded (flat) $C^2$ curved-crease arc is denoted by the parametric function $\ell(s)$, where $s$ corresponds to the $x$-coordinate of the crease

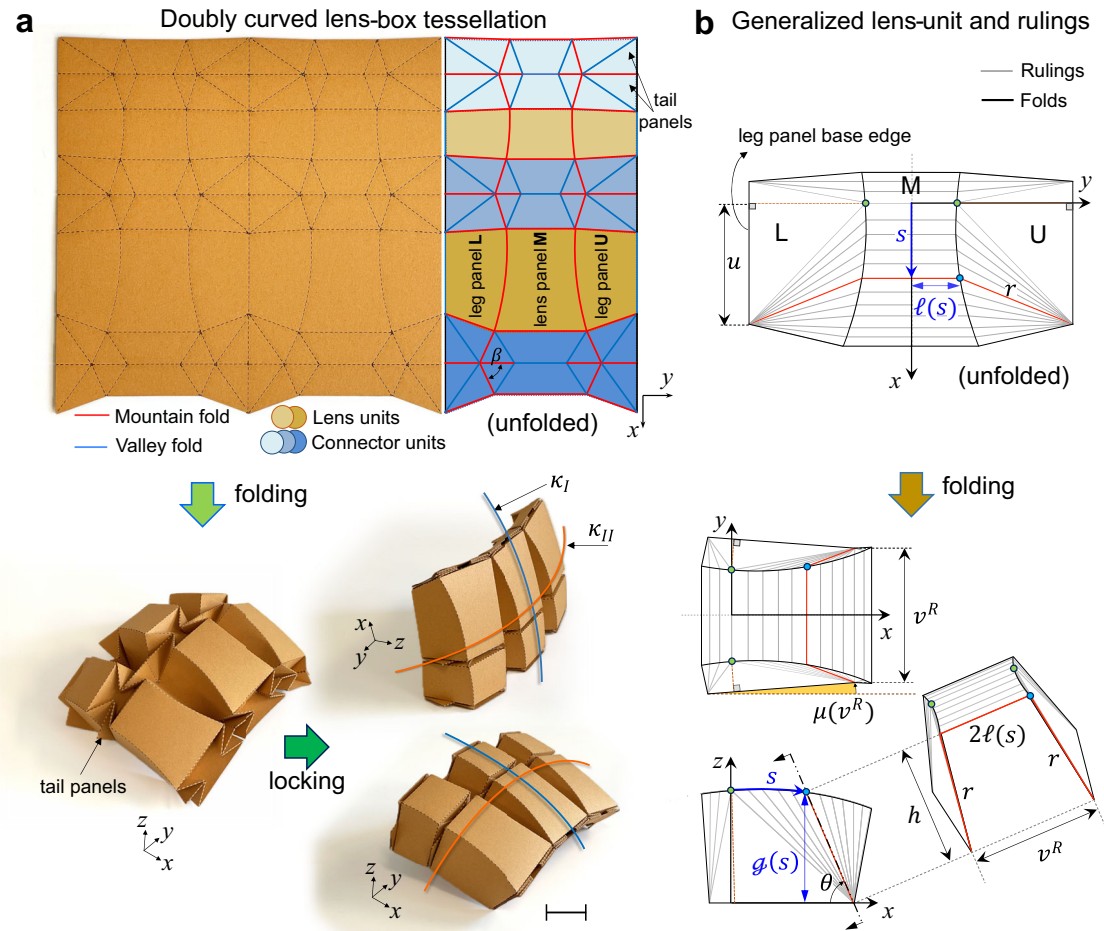

**Fig. 1 | Doubly curved lens-box tessellation and geometrical construction of the generalized lens-unit. a** A 2 × 3 tessellation (2/3 in orange paperboard on left and corresponding crease specification on right) of non-symmetric (with respect to the *y*-axis) doubly curved lens-box units, specifying mountain and valley assignments and highlighting (with given shades) the dissimilar connector and lens units. The paperboard prototype below illustrates its partially folded state (bottom-left) and two views of the locked state (bottom-right), which conforms to a doubly curved surface with smooth curvature $\kappa_I$ (blue curve) and piecewise curvature $\kappa_{II}$ (orange curve). **b** Representative generalized lens-unit in its unfolded (developed) and folded state. (top) shows a valid rigid-ruling foldable ruling segmentation, with parametric curved-crease function $\ell(s)$ and a generic curved-crease point *s* with

highlighted rulings with thick red lines: the cone ruling ending at crease point *s* has the length $r = r(s)$. (bottom) illustrates a partially folded lens-unit from top view, side view, and sectional view, specifying its main folding geometric parameters. In the folded state, the specified thick red rule lines, together with the line connecting the free edges of the red rulings, form a planar trapezoid with height $h = h(s)$ and base length of $v^R$. The 2D parametric function $\mathscr{g}(s)$ represents the orthogonal projection of folded panel M with respect to the *x-z* plane and can be expressed in a polar coordinate as $(h(s), \theta(s))$. Our shown ruled surface idealization excludes the effect of the free-boundary leg panels, such as those apparent in (**a**)-bottom, which deform differently. Scale bar, 25 mm.

point (blue dot in Fig. 1b). We introduce the folding parameter $v^R$, which measures the distance between the apices of the right cone-ruled segments of the U and L panels in the folded state (Fig. 1b-bottom), and denote the *x*-coordinate of the corresponding cone apices in the unfolded state as *u*. Let $\mu(v^R)$ denote the angle between the base edge of the lower leg panel, treated as a directed vector from its left endpoint to its right endpoint, and the positive *x*-axis in the folded state, measured counterclockwise as positive. Denoting the orthogonal projection of the folded panel M onto the *x-z* plane by the parametric function $\mathscr{g}(s)$—where *s* now represents the arc length—we derive the fundamental geometric constraint that governs the smooth folding of the curved crease (see Supplementary Note 2.1.1) as

$$\frac{\ell'(0) + \left(\frac{v^R}{2} - \ell(0) - u\sin\mu\right)\mathscr{g}'(0)/\mathscr{g}(0)\cos^2\mu}{\sqrt{1 - \mathscr{g}'(0)^2}} + \tan\mu = 0 \qquad (1)$$

Furthermore, $\mathscr{g}(s)$ can be explicitly expressed as a function of the unfolded crease $\ell(s)$ (see Supplementary Note 2.1.1), allowing the

entire folded geometry to be described through a single independent folding parameter ($v^R$). Hence, the lens unit exhibits a single degree of freedom (DOF).

## Rigid-ruling foldability and locking

Having described the geometry of our 1-DOF rigid-ruling foldable lens unit, we now discuss the folding of the connector and the entire pattern (see Supplementary Note 2.2). We show that the connector is also rigid-foldable with 3-DOF, similar to the original waterbomb pattern[55,56] (see Supplementary Note 3). This can reduce to either 2-DOF with symmetric folding or 1-DOF when its tail panels come into contact at a specific folding state (Fig. 1a-bottom and Supplementary Note 3.3, Supplementary Fig. 9). Once the connector flat-folds fully, the lens unit can no longer fold as it has reached a locked state (Fig. 1a-bottom). While local flat-foldability can be verified using *Kawasaki's*[57] and *Maekawa's* Theorems[58,59], these are insufficient for assessing global flat-foldability in our 4-vertex connector unit. To design a flat-foldable connector, we geometrically construct its flat-folded state and derive constraints that prevent edge and vertex penetration (see

Supplementary Note 2.2.1). Satisfying these flat-foldability conditions also ensures its full-range rigid folding motion.

Assessing the rigid-ruling foldability of a multi-vertex pattern such as our doubly curved lens-box tessellation is NP-hard[60]. Here, we can only prove the rigid-ruling foldability of a single doubly curved lens-box unit and its unidirectional tessellations (see Supplementary Note 3.4). In particular, we use Freeform[61] rigid origami simulations to demonstrate that our pattern deforms isometrically and remains rigid-ruling foldable once tessellated in plane (see Supplementary Note 3.5). To achieve this, we first approximate the curved-crease fold pattern with a piecewise linear version consisting of a finite number of ruling segments, forming a discrete ruled surface. While the mobility of the resulting pattern remains independent of the ruling discretization, its folded geometry converges to the smoothly folded curved-crease shape as the number of ruling segments increases. Although we cannot fully analyze the folding mechanics of the tessellated pattern through rigid-folding simulations alone, we observe that in practice, folding becomes more challenging when the sector angle $\beta < \pi/2$ (Fig. 1a-top), and the folding difficulty increases as $\beta$ decreases. We attribute this phenomenon to the kinematics of the origami pattern, where some creases exhibit a non-monotonic evolution of their dihedral angles and sharply reverse their folding direction. While this behavior is manageable for a single unit, it becomes strongly amplified when several connector units are tessellated and required to fold compatibly (see Supplementary Note 3.3.2 and Supplementary Fig. 9).

## Locking into smoothly curved surfaces

With the geometric construction described above, we now address a more interesting question: Given a smooth 3D surface $\Omega$, can we find a 2D doubly curved lens-box tessellation that, upon locking, can be isometrically embedded onto $\Omega$ while maintaining $C^1$ or $C^2$ smoothness along one principal direction? This is equivalent to finding a locked pattern where the panel M conforms to a prescribed meridional curvature $\kappa_I = 1/R_{GI} \equiv 1/R_c$ (the curvature of $\mathcal{G}$, shown as the blue curve in Fig. 1a-bottom), while its tessellation along the azimuthal direction approximates the given curvature $\kappa_{II} = 1/R_{GII}$ (orange curve in Fig. 1a-bottom) in a piecewise linear manner (see Supplementary Notes 4 and 5). This problem is explicitly formulated as an inverse problem that can be solved through constrained optimization (see Supplementary Note 6). Constraints for the lens units involve smooth folding of the crease and $C^1$ smoothness along the meridional direction for two adjacent units (see Supplementary Note 4.1). For the connectors, constraints include achieving flat-foldability without overlap or collision, and preventing panel protrusion beyond the target surface (see Supplementary Note 2.2.1). Additionally, the effect of panel thickness is considered when matching the meridional curvature, as flat-folded connectors typically have a non-negligible thickness of $\sim 8t$.

While our formulation can determine planar tessellations that conform to double-curvature surfaces with uniform curvature (see Fig. 1a), folding these patterns remains challenging, and convergence is not always guaranteed. Moreover, for variable-curvature surfaces, obtaining such tessellations is generally impossible. We develop here an alternative strategy to tackle these problems: an explicit, additive tessellation approach that resembles a layer-by-layer 3D printing fabrication (see Supplementary Note 5.2). We begin by partitioning the generator curve of the given surface into sections of constant curvature. Initially, we construct a layer of the locked units that matches the specified surface curvatures, $\kappa_I$ and $\kappa_{II}$. Subsequent layers are then independently constructed using the geometry established in the preceding layer. As a result, the unfolded crease patterns for each layer remain separate (disconnected) but can be connected to neighboring layers along the meridional direction once locked. In contrast to double-curvature surfaces, cylindrical surfaces can be fabricated by folding a single, continuous sheet of paper with a tessellated pattern,

whose unit cells remain seamlessly connected throughout the entire folding process, from the unfolded to the locked state (see Supplementary Fig. 13a, b, and Supplementary Note 5.1). We showcase the versatility of our approach with closed-form relations enabling constant- and varying-curvature cylindrical surfaces, including a closing ring, logarithmic spiral, and a chair-like form (Fig. 2a, b, c), as well as other non-$C^1$ periodic tessellations (see Supplementary Note 5.1.4). Figure 2a-g shows the results of our analysis applied to a geometrically diverse set of cylindrical and doubly-curved surfaces, with paperboard prototypes fabricated through laser perforation and manual folding (see Methods). Cylindrical surfaces are folded from a single sheet (no adhesive), whereas doubly curved surfaces are assembled by folding strips into layers in the $x$–$y$ plane and bonding along the $z$-direction.

Scaling the origami pattern enables to determine its curvature design space; however, in practice, the attainable curvature is bounded by thin-panel criteria on the base sheet (e.g., $t/\ell_{\text{crease}} \lesssim 0.01 - 0.05$, $t/R_c \lesssim 0.01$, where $\ell_{\text{crease}}$ is a characteristic crease length). Achieving larger curvature (smaller $R_c$) generally requires finer tessellations and thinner base material, which in turn reduces the global thickness of the folded shell.

## Embedding tendons into origami units

Our doubly curved lens-box unit exhibits multi-DOFs before locking. Once locked, it becomes stiff under uniform in-plane compression (Fig. 3a top-right). However, upon tessellation into a periodic shell, the overall structure's DOFs increase, creating internal mechanisms. In practice, additional non-rigid deformation modes may also emerge due to the panel flexibility (Fig. 3b). While some origami tessellations[30,33], such as our lockable lens-box pattern without tendons (see Fig. 3c), can achieve limited stiffness or maintain partial rigidity under specific compressive loads and boundary conditions, achieving global rigidity in existing origami tessellations made of compliant panels remains a significant challenge.

Here, we postulate that the addition of strategically oriented tensile members, namely tendons (Fig. 3c), to the doubly curved lens-box units can suppress all DOFs of the system, ensuring multiaxial rigidity regardless of the direction of the applied loads and boundary conditions. Since rigidity under compression is achieved through locking, the added members must only warrant rigidity under tension: a harmony between tensile and compressive members reminiscent of tensegrity structures[62–64]. As illustrated in Fig. 3c, these tendons pass through the top-middle and bottom corners of the flat-folded connector units. By stretching the tendons, the partially folded units flex towards their locked state (Fig. 3c-right). When the connectors flat-fold, the tendons reach their shortest length, and the locked units squeeze their neighboring units. Pre-tension in the tendons acts analogously to applying in-plane pressure on individual building blocks; it enables preserving the integrity of the entire shell structure regardless of any external compressive force.

We show that the addition of tendons not only prevents deformations arising from multiple rigid-ruling folding motions, but can also restrain the non-rigid-ruling deformation modes (e.g., Fig. 3a, b) upon strategic tendon arrangement. For example, the paperboard curved structure in Fig. 3d shows nearly zero deflection against a twisting action when rigidified using the tendons, unlike its glued counterpart, which severely deforms under identical conditions. To determine the tendon position and layout illustrated in Fig. 3c, d, we resort to structural rigidity analysis and convert our locked unit into an equivalent triangulated network of bars and joints (see Supplementary Note 7). The bottom tendons are self-stressed, a feature advantageous for imparting first-order stiffness to the frame[63,64].

Our model aims to capture the structural rigidity of the doubly curved lens-box solely in its locked state, where mobility reduces to zero under compression. Here, we assume that our bent panels,

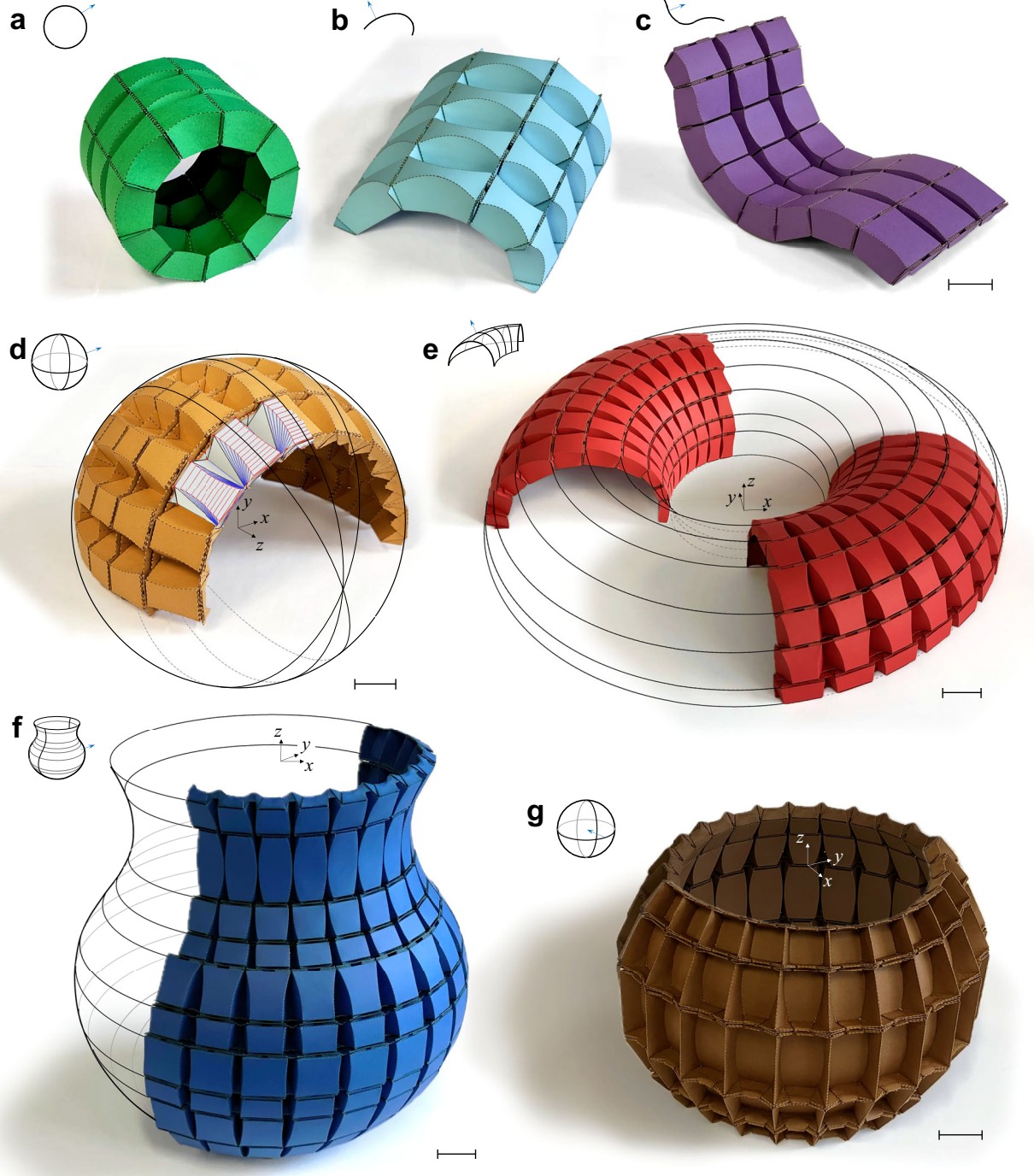

**Fig. 2 | Physical paperboard origami tessellations for prescribed curved surfaces after locking.** Representation of inversely designed smoothly-curved cylindrical structures (**a**–**c**), and smoothly-curved (along meridional direction) double-curvature structures (**d**–**g**) in their locked state. Schematics in the top-left corner show the prescribed target geometry besides the normal vector of the front side of the folded paper. **a** Constant curvature cylinder−closing ring. **b** Logarithmic spiral. **c** A variable curvature cylindrical surface−chair. **d** A portion of a convex sphere−positive Gauss curvature with $\kappa_I > 0$, $\kappa_{II} > 0$−with the superimposed rigid-folding simulation results for only three folded cells, obtained using Freeform origami software[61]. **e** Portion of a torus−a combination of positive and negative Gauss curvatures (negative portion has $\kappa_I > 0$, $\kappa_{II} < 0$). **f** Half of an ancient vase with a base curve (generator) comprising three distinct curvature radii−a combination of positive and negative Gauss curvature (negative portion has $\kappa_I < 0$, $\kappa_{II} > 0$). **g** A portion of a concave sphere (internally smooth)−positive Gauss curvature with $\kappa_I < 0$, $\kappa_{II} < 0$−illustrating an example of the internal backbone structure with architecture imparting load-bearing capacity. In (**d**–**g**), locked origami layers fabricated additively by folding individual layers and stacking along the $z$-axis. In panels (**a**–**c**), the origami patterns are folded from a single sheet. In panels (**d**–**g**), the patterns are assembled by folding strips into layers in the $x$-$y$ plane, which are then glued along the $z$-direction to form the complete surface. To maintain the shell structures in their locked configuration, adjacent panels of connectors are bonded. Scale bar in (**a**), (**b**) and (**c**) = 25 mm, in (**e**) = 35 mm, and in (**d**), (**f**) and (**g**) = 20 mm.

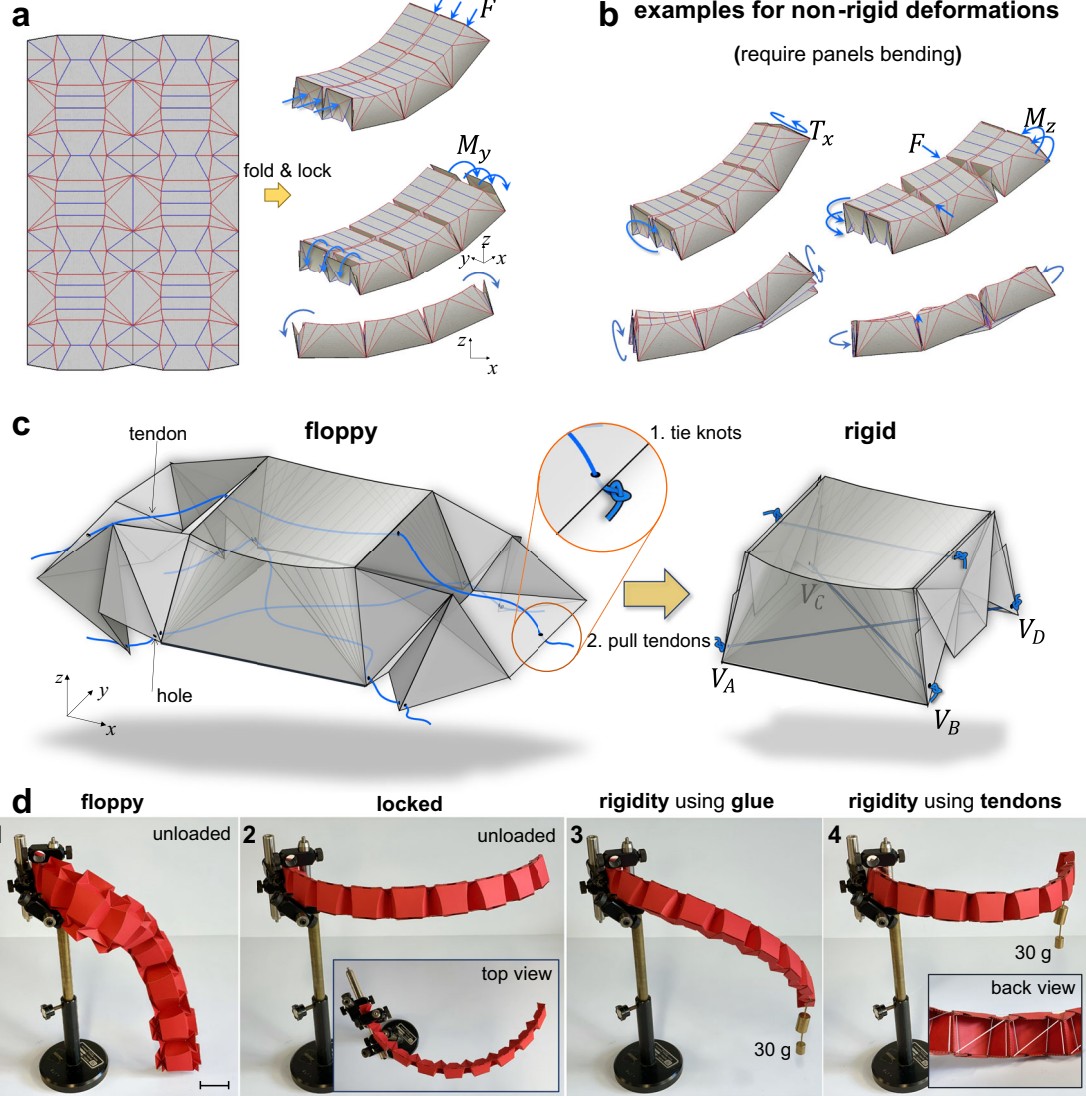

**Fig. 3 | Rigid-ruling and non-rigid-ruling deformation modes, and design concept for integrating tendons into our origami building blocks. a** A representative doubly curved lens-box tessellation pattern in unfolded state (left) along with its locked configuration (right-top), which maintains its integrity through compressive force $F$ applied along the slightly curved locked shell. It, however, loses its rigidity once subjected to a bending moment. **b** Possible non-rigid-ruling deformation modes of the locked shell subjected to non-uniform or twisting loads. **c** Integration of the tendons (blue) in our partially folded origami unit inspired by tensegrity notions. Upon tendon stretching, achieved in this schematic through tendon pull and then knot tie, the unit maintains its locked state without the application of external constraints and only through the internal tendon mechanism. While the top tendon is sufficient to restrain the rigid-ruling folding motions, the set of bottom tendons can confine the non-rigid-ruling deformation modes (infinitesimal modes) of the locked structure. **d** A paperboard prototype of a unidirectionally tessellated doubly curved lens-box (which upon locking conforms to the third external layer of the torus in Fig. 2e) when constrained from one end and subjected to four conditions: (1) Soft partially folded state which is saggy under its own weight; (2) Stiff state achieved by gluing the connector panels in their flat-folded state subjected to its own weight (~5 g); (3) Locked origami structure (2) revealing a large twisting deflection under load; (4) Origami structure (1), locked by the tendon pull, showing zero deflection achieved by adjusting pre-tension to counteract the applied load; inset illustrates tendon arrangement. Scale bar, 30 mm.

defined by the ruling segmentation, have zero flexural stiffness, and the rulings are modeled as creases with zero torsional stiffness. This analysis examines the inextensional deformation modes—finite and infinitesimal DOFs—that our locked origami may possess. To this end, we made the following assumptions: (i) the creases do not bend, following the asymptotic analysis of the *Föppl–von Kármán* equations which show that, at the thin panel limit, the energy required to bend a strip of straight crease is five times larger than the energy required to stretch it[65,66]; (ii) the connector unit panels in the flat-folded state resist bending due to their four- or eight-layer thickness, making bending energetically less favorable compared to that of single-layer panels in the lens unit.

While our simplified truss-based model qualitatively demonstrates the rigidity of our tendon-origami structure, it is not suitable for analyzing its mechanical response. A more advanced computational model—accounting for panel bending stiffness and interfacial contacts between connector panels, as well as geometric, material, and contact nonlinearities—would be required to fully unveil the underlying mechanics and failure mechanisms. This is beyond the scope of this paper, the focus of ongoing research.

## In situ tunable load-bearing capacity
To precisely adjust the flexural rigidity of our tendon-origami shells (Fig. 4a), we incorporate a simple one-way rotational gear mechanism

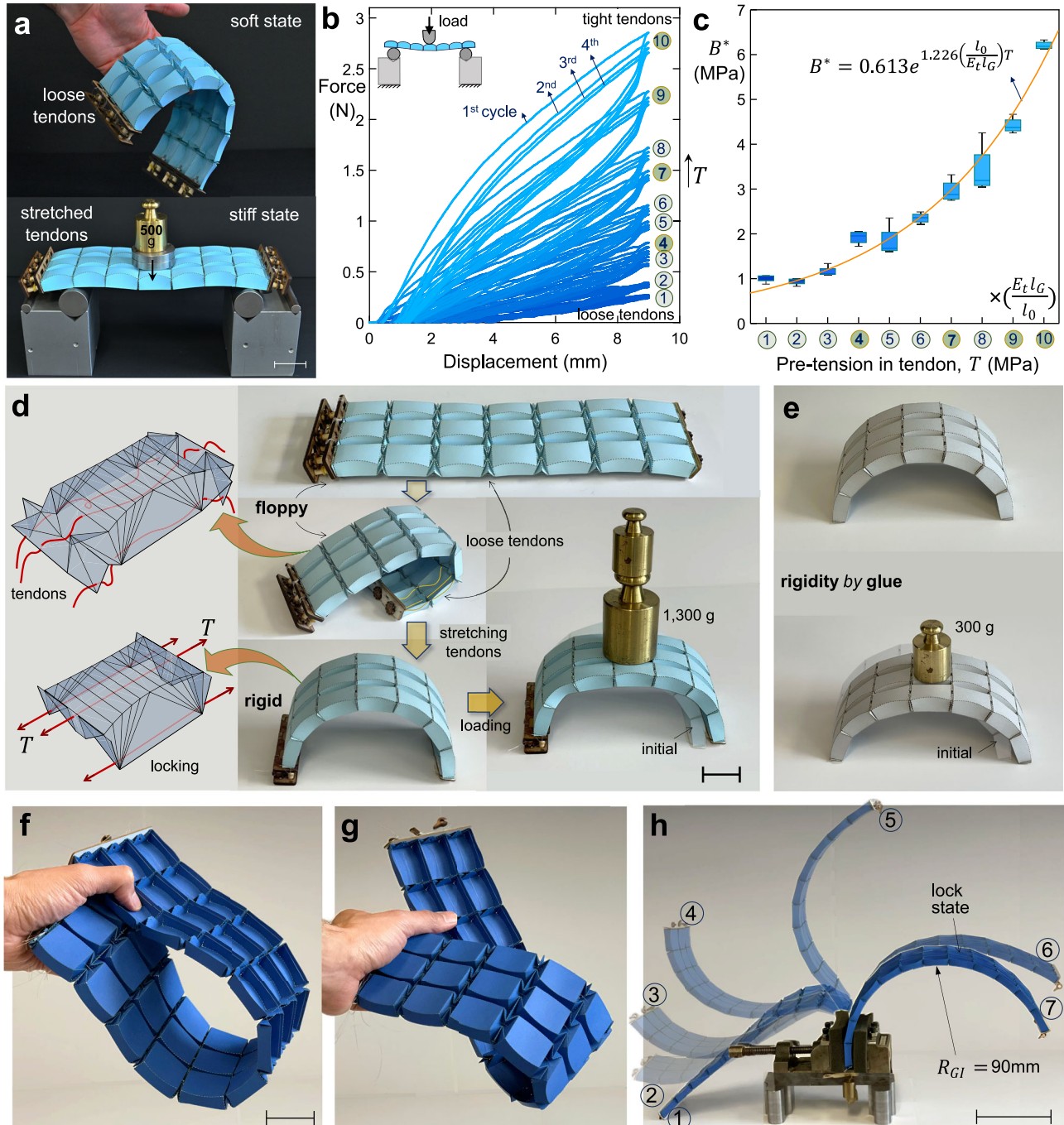

**Fig. 4 | Load-bearing properties, reprogrammable stiffness, and actuation of a tunable tendon-origami shell. a** Paperboard prototype in soft mode (top) demonstrates flexibility under loose tendons, and in stiff mode (bottom) functions as a load-bearing horizontal bridge after tendons tightening via integrated gear mechanisms. **b** Experimental force-displacement curves obtained through three-point bending tests (four cycles of loading-unloading), showing incremental change in material properties by increasing tendon pre-tension for ten increments denoted with circled numbers. While the lower tendons are equally stretched in each increment, the upper tendons are only stretched during the increments 4,7,9 and 10 (marked with darker shades) to flatten the unloaded geometry of the shell. The magnitude of each increment depends on the tuning gear mechanism, and here specified with $l_G$. If $l_0$ denotes the initial length of the tendon, and $E_t$ its Young's modulus, the pre-tension $T_i$ in each increment is $T_i = E_t l_G / l_0$. **c** Apparent elastic flexural modulus $B^* = KL^3/4Wh_0^{*3}$ vs. tendon pre-tension $T$, depicted by box

plots from our four testing cycles. Sample dimensions (in mm) are $L = 188$, $W = 82$, and $h_0^* = 12.5$. **d** (left) Schematic of a doubly curved lens-box unit representing arrangement of applied tendons in tunable tendon-origami shells, upon loose tendons (highlighted with yellow for two cells in the middle panel) and locking. (right) Fabricated paperboard prototype (only the paper portion weighs 8 g) of our deployable shell shown in two relaxed states and also upon transforming into rigid smoothly curved form, achieving a load-to-weight ratio of 162 without external support. **e** Non-deployable arch counterpart (for (**d**)) rigidified by gluing its connector panels with reduced load tolerance. Highly reconfigurable tendon-origami shell prototype, in two soft distinct configurations (**f**) and (**g**), along with actuation (**h**) into various stable configurations and deployment stages from loose into a predefined arc. Scalebar in (**d**) and (**e**) = 30 mm, in (**a**), (**f**) and (**g**) = 35 mm, and in (**h**) = 80 mm. Source data for panels **b** and **c** are provided as a Source Data file.

designed to stretch individual tendons incrementally (see Methods and Supplementary Note 8). By pulling the tendons, the partially folded soft shell gradually reconfigures into its stiff (locked) state with zero-DOF (Supplementary Movies 1 and 2). In this configuration, the tendon-origami shell internally jams, pressing the adjacent building blocks with a force proportional to the tendon stretch (Fig. 4a). This process, which to some extent recalls jamming in granular materials, inhibits rigid-ruling and non-rigid-ruling deformation modes. While tendons experience stretching, our lens units tolerate compressive loads in harmony, gradually transitioning from a soft to a stiff state. This mode-shift can reversibly, continuously, and quickly enhance the flexural modulus of our tendon-origami shells in operando.

We quantify changes in mechanical properties as a function of the tendon pre-tension $T$ by performing a series of three-point bending experiments (see Methods and Supplementary Note 8) and calculating the apparent elastic bending modulus $B^*$ of the shell (Fig. 4b). Stretching a tendon with initial length $l_0$ and Young's modulus $E_t$ by $l_G$ creates the pre-tension $T = E_t l_G / l_0$ (see Supplementary Note 8.2). If the average stiffness $K$ of the initial elastic regime (Fig. 4b), and the initial planar dimensions $L$, $W$, and the thickness $h_0^*$ of the shell are known, then $B^* = \frac{KL^3}{4Wh_0^{*3}}$. Our results in Fig. 4c show that the apparent flexural modulus increases convexly with the applied tendon pre-tension due to the (i) in-plane-bending coupling (geometric/stress stiffening) in the pre-stressed shell and (ii) staggered engagement of the top tendons at steps 4, 7, 9, and 10 to re-flatten the unloaded shape. This nonuniform engagement causes the effective pre-stress (top and bottom) to rise nonlinearly with the nominal increment in the bottom tendon pre-tension, yielding an exponential-like trend in the bending stiffness. We terminated the loading at the final reported point, where additional pre-tension triggered cell distortion and local buckling. Beyond this onset of instability, further pre-tension did not translate into a reliable stiffness increase for this configuration. (The lowest modulus value remained unregistered due to the challenge of measuring the initial tendon length in a formless relaxed state).

Figure 4d illustrates our paper-based arched deployable shell, exhibiting a remarkable load-to-weight ratio (approximately 162), surpassing that of the glued (rigid) counterpart (see Fig. 4e) by a factor of 4. During actuation, the tendon-origami shell can attain various configurations depending on its initial state and external loads. For instance, Fig. 4f–h shows the shell actuating under its own weight (14 g), can transition in multiple stable forms before locking into its prescribed load-bearing arc. In Fig. 4f–g, the tendon-origami shell is stabilized by panel bending and unit self-contact under a moderate boundary constraint at the edges (manually held during the demonstration). In Fig. 4h, the shell attains free-standing equilibrium states (① - ⑦) where gravity is balanced by partial, non-uniform tendon pre-tension, panel bending rigidity, and unit contact; a full pre-tension of the tendons produces the final locked configuration (⑦).

Tendons enable precise control over the flexible shell kinematics, facilitating a wide range of motions and configurations, particularly suitable for soft robotics applications in constrained environments, e.g., endoscopic surgery. Unlike existing metamaterials with continuously tunable modulus[9,67,68] that rely on high temperatures[67], electrical/magnetic fields[68], or pneumatic pressure[9]—posing risks during use or failure—our design remains safe and functional even in the event of partial failure. This is primarily because our tendon-origami system relies on lockable building blocks with multiple independent tendons, enabling load redistribution upon failure of a few tendons. As a result, partial failure does not compromise the overall integrity or functionality of the system. In contrast, systems controlled by pressurized fluids can fail catastrophically upon puncture, whereas tendon-origami mechanisms remain robust, typically unaffected by the failure of a single tendon, offering a reliable approach for programming stiffness and motion in a robotic system.

## Discussion

This work has introduced a hybrid origami tessellation combining straight and curved creases, capable of isometric folding and locking into structural shells with smooth doubly-curved or cylindrical surfaces with varying curvature, thereby addressing the trade-off between flexural rigidity and curvature smoothness in traditional tendon-free origami tessellations (see Fig. 5). We have explicitly formulated an inverse design problem whose solutions can generate building blocks for the deployment of load-bearing structures with complex geometries. The integration of tendons has enabled our origami shells—unlike existing deployable origami concepts, which are typically compliant or, if lockable, require structural support to tolerate mechanical loads—to transform gradually and reversibly from soft to multiaxially rigid, load-bearing shells with tunable stiffness. Paperboard prototypes confirm the existence of smoothly curved shells and their mechanical characteristics. Demonstrated as rigid-ruling foldable, the discretized, ruled segmentation version of our patterns can be adapted to non-deformable materials or thick-panel origami.

While our previous work[42] introduced a symmetric hybrid crease tessellation capable of forming cylindrical surfaces with constant curvature, the present framework establishes a distinct, non-symmetric, doubly curved lens-box pattern with fundamentally dissimilar geometric construction, folding kinematics, and actuation mechanisms. These differences yield markedly dissimilar physical outcomes and mechanical functionalities—most notably, the ability to form smooth, doubly curved, load-bearing shells with variable curvature and continuously tunable stiffness—which are inaccessible in our earlier work.

Our formulation and doubly curved lens-box crease pattern can be generalized by breaking the symmetry in the lens unit of the unfolded pattern, allowing for folding into non-symmetric surfaces with spatially varying Gaussian curvature. Moreover, by orienting the waterbomb units in multiple directions, more complex versions of the lens-unit can be realized, paving the way for a wider variety of generalized doubly curved surfaces. We emphasize that any structure folded from a developable sheet can only approximate curvature in one direction of a doubly-

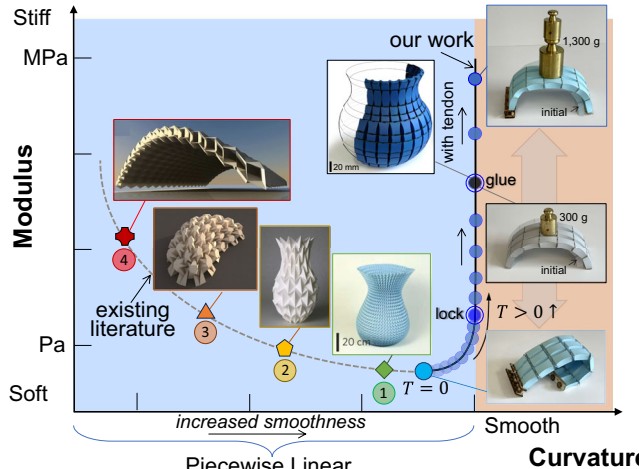

**Fig. 5 | Trade-off between rigidity and curvature smoothness.** Qualitative rigidity-smoothness plot for surfaces obtained from existing tendon-free origami tessellations, illustrating the trade-off between flexural rigidity and curvature smoothness. The plot compares our adaptive origami shells with smoothly curved structural surfaces with benchmarks of origami concepts that approximate curved surfaces and cannot modulate their stiffness. The inset images include paperboard models of our work and two literature examples (inset 1 reproduced from ref. 20 and inset 2 reproduced from ref. 70.) under the Creative Commons CC-BY-NC-ND 4.0 license (https://creativecommons.org/licenses/by-nc-nd/4.0/); no changes were made. Insets 3 and 4 are computational renders or models adapted from refs. 25,33, respectively, and reproduced with permission.

curved surface with nonzero Gaussian curvature. This is an inherent limitation of using origami to fold flat sheets into doubly-curved surfaces.

Our design principles, stemming from origami and tensegrity notions, are inherently material-agnostic and scalable, yet not scale-independent, enabling the use of various base materials (e.g., metals, polymers, composite carbon fiber sheets) and the scaling of the crease pattern to suit diverse applications, from space missions to safety helmets. At larger scales, factors such as material properties, gravitational forces, geometric and material imperfections due to manufacturing might become dominant, influencing mechanical properties and failure mechanisms. The upscaling and downscaling of our concept would thus require further investigations.

By leveraging the synergy between lockable hybrid crease origami and tensegrity notions, our approach opens avenues for the design of deployable and adaptive load-bearing curved structures, including wearable exoskeletons, temporary emergency tents, reconfigurable aerofoils, haptic architectures, morphing robots, and smart fabrics.

## Methods

### Materials and fabrication
We fabricated all origami prototypes from 0.21-mm-thick cellulose paperboard sheet (200 gm$^{-2}$ Fabriano Craft paper) by laser-perforating flat sheets along the prescribed crease pattern (CM1290 laser cutter, SignCut Inc.) and then manually folding them. Fold lines were produced using 1.2-mm-long cuts, uniformly spaced at 1-mm intervals. To hold the tendon-free origami shells in their locked configurations, adjacent facets of the waterbomb connector units were glued in their flat-folded state using a commercial polyvinyl acetate (PVA) paper adhesive. Sample manufacturing, including laser-cutting, folding, and tendon implementation, was conducted at ambient temperature (22 -23 °C) and a relative humidity of ~ 30-40%. The tensile Young's modulus and strength of our paperboard sheets were measured to be approximately 7.9 GPa and 57 MPa, respectively, in the machine direction (MD), which are roughly twice the values measured in the cross direction (CD)[30].

The gear-based tuning mechanism were made of high-density wood fiberboard (Masonite Panel) with a thickness of 3 mm for all planar parts, such as gears, stoppers, and panels, along with hardwood dowel rods with a diameter of 5 mm and a length of 20 mm for the shafts (see Supplementary Notes 8.2 and Supplementary Fig. 23). Tendons are made of monofilament nylon fishing line (Red Wolf, Canadian Tire Corporation, Canada) with a nominal load capacity of 3.6 kg and a diameter of about 0.1 mm. While we did not perform experiments for measuring the properties of the fishing line, its Young's modulus varies between 1.5 and 2.4 GPa, and its ultimate strength between 0.25 and 0.9 GPa[69].

### Mechanical testing
Experiments were conducted using an electrodynamic mechanical testing equipment (StepLab, STEP Engineering S.r.L., Resana, Treviso, Italy) equipped with a 1kN load cell (AEP Transducers, Cognento, Italy; Type TSC3) and operated under displacement-control conditions. To maintain quasi-static loading condition, we applied a displacement ramp corresponding to a nominal strain rate of $10^{-3}s^{-1}$. Consistent with the fabrication procedure, all tests were performed at ambient temperature and a relative humidity of around 30–40%. For three-point bending experiment, specimens were supported on two cylindrical rollers (30 mm diameter), and loading was applied by lowering an indenter with a smooth cylindrical steel head of the same diameter. Displacement was obtained from the crosshead readings, which provide sufficient accuracy for our purpose, as discussed previously[30].

### Tendon pre-stress tuning mechanism
To tune the tendon prestress, we fabricated a gear-based tuning mechanism (Supplementary Note 8.2). The mechanism enables controlled pretensioning and can be locked after applying a prescribed tendon stretch to maintain the applied prestress and the locked shell configuration. The tuning mechanism was attached to the origami shell in the locked state using three screws (Supplementary Fig. 23c). The gears permit rotation in one direction only, with reverse motion prevented by stoppers. A rubber band was used to pull the stopper towards the gear shaft, thereby conditioning the stopper's motion. Gears and stoppers were bonded to their shafts, allowing the shafts to rotate freely within the mounting holes. Tendons were secured by passing their ends through a small transverse hole drilled in each shaft.

### Construction of the 'Doubly Curved Lens-Box' Pattern
The doubly curved lens-box unit combines two folding motifs: a lens-unit and a variation of the waterbomb. Its geometry consists of one lens unit and two non-uniform waterbomb connector units. To analyze the overall folding kinematics, we study these components separately, starting with the lens unit. During folding, we assume that all free boundaries (edges) of the lens unit remain undeformed (rigid) and that the straight creases of the waterbomb connectors do not bend. The detailed characterization of our doubly curved lens-box pattern, including its ruling segmentation and geometric construction, is presented in Supplementary Notes 1 and 2.

### Rigid-ruling foldability
Rigid-foldability describes the property of an origami pattern to fold through the sole rotation of its creases without facet deformation. The proof of rigid-ruling foldability of our doubly curved lens-box pattern and related discussion are provided in Supplementary Note 3.

### Tessellation generation for target surface mapping
The necessary conditions and constraints for smooth folding of the lens unit, flat foldability of the waterbomb connector units, smooth connection of the lens-box units upon tessellation, and the general rigid-ruling foldability of the entire crease pattern are provided in Supplementary Notes 1–4. Based on these relations, we identify tessellations that match the first (global) principal curvature $\kappa_I$ of the target surface along the lens panel, i.e., arc-length $s$, and approximate the second principal curvature $\kappa_{II}$. Supplementary Note 5 formulates the construction of generalized cylindrical surfaces with $\kappa_{II} = 0$, and Supplementary Note 6 presents a numerical optimization procedure to search for locked lens-box embeddings of intrinsically curved (non-zero Gaussian curvature) target surfaces. We denote the radii associated with the first and second global principal curvatures by $R_{GI}$ and $R_{GII}$, respectively, and use $R_c$ for the local radius of the curvature of the lens panel M.

### Rigidity analysis
We analyze the rigidity of the origami shell structure under the following assumptions. First, the curved-crease unit is converted into an equivalent rigid-origami model defined by its ruling pattern (Supplementary Fig. 21). A diagonal crease is added (introduced only in the rigidity model) to the quadrilateral panels of the lens panel to allow twisting deformation modes. Second, all panels are assumed to be infinitely rigid, and the rulings and crease lines are modeled as frictionless rotational hinges. Third, to simplify the formulation, we replace the rigid origami structure with a triangulated network of inextensible struts connected through pin joints. Additional details are provided in Supplementary Note 7.

## Data availability
All data needed to evaluate the conclusions in the paper are present in the main text, the Supplementary Information, and the other supplementary files. Source data are provided with this paper.

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

## Acknowledgements

This work has been supported by the Canada Research Chairs program (Grant No. 257679), Natural Sciences and Engineering Research Council of Canada (Grant No. 208241), McGill Engineering Doctoral Award (MEDA), and Fonds de recherche du Québec – Nature et technologies (FRQNT).

## Author contributions

Conceptualization, visualization: M.M.; research design, investigation, writing—original draft and writing—review & editing: M.M. and D.P.; methodology: M.M.; supervision: D.P.

## Competing interests

Patent applications related to this work have been filed: Canadian patent application (CA 3,289,886) and U.S. patent application (US 19/368,610). The authors declare no other competing interests.
