## [Transparent Peer Review file · Nature Communications]

Smooth doubly curved origami shells with reprogrammable rigidity

Corresponding Author: Professor Damiano Pasini

Version 0:

Reviewer comments:

Reviewer #1

(Remarks to the Author)

This manuscript presents a new origami structure called the doubly curved lens-box, which combines curved and straight folds. By embedding pre-tensioned tendons, the system enables reversible transformations from soft, compliant states to rigid, load-bearing structures with tunable stiffness. The authors present an inverse design framework for generating tessellations that conform to prescribed curved surfaces and validate the concept with paperboard prototypes, showing potential for applications in fields such as exoskeletons, deployable aerospace structures, robotics, and adaptive architectural systems.

The idea is interesting and the presentation is generally clear. I have some suggestions for the authors to improve the manuscript:

1. It is strongly suggested that the authors stick to the terminology of "rigid-ruling foldability," rather than using rigid foldability and rigid-ruling foldability interchangeably, because by the most well-accepted definition, curved origami cannot be rigid foldable because it necessarily involves panel bending.
2. Line 151-152: this claim is wrong: "This property has significant implications as it enables the use of even non-deformable materials to create rigid-foldable origami structures..." Again, curved origami necessarily involves panel bending – no matter the amount, which is impossible for strictly non-deformable materials to achieve. Therefore, a softer tone is needed, such as saying "even stiff materials." This is a problem throughout the manuscript. The authors have to be very careful of making strong claims as the proposed designs are not strictly "rigid origami."
3. The authors mentioned about difficulties during folding, and indicated that some geometric parameters lead to deviation from rigid-ruling foldability. It is then important to investigate such phenomenon by conducting FEA, or bar-and-hinge model analysis, at least on a few particular patterns as preliminary results. It is not ok just saying such critical study is out-of-scope, when this matters to the claims in this manuscript.
4. On a related note, in the constrained optimization, the authors should report how much the constraints are violated in the end, which is very likely, as in Ref. 20.
5. Line 246-247: It should be clearly indicated in Fig. 2 about whether a pattern is folded from an entire piece or glued from many separate parts. The cut-and-glue strategy is fine, but it is important to clearly show this to the readers.
6. All demonstrations in this work are made of paperboards. However, paperboards are quite forgiving for stretching deformation, which, by theory, should not present in this work. Therefore, it is important to show material versatility of the proposed designs by trying a different material, especially materials with large in-plane stiffness.
7. There are very little explanations for Fig. 4F-H. Please elaborate on how the different configurations are stabilized.

Reviewer #2

(Remarks to the Author)

This paper presents a new origami folding pattern -- called "doubly curved lens-box" that can develop a flat sheet precursor into a doubly curved surface (with a smooth curvature and piecewise curvature). This is a significant step forward from the author's previous paper, which showed only a single curved surface. In addition, the authors proposed a novel method of locking the folded curvature with embedded tendons, demonstrating the practical use of this new origami pattern for different

engineering functions.

Overall, this paper presents an impressively thorough framework for the design and kinematic analysis of the doubly curved lens-box. The tendon locking is a nice addition. Therefore, I recommend publication with the following suggestions for further improvement.

Reachable design space: Although the authors have demonstrated several fascinating examples in Fig. 2, it is never entirely clear what range of curvatures the doubly curved lens-box can achieve. Is there a theoretical limit on the obtainable κ_I and κ_{II} ?

Rigid foldability: Is local rigid foldability only a necessary condition for global rigid foldability? That is, have you encountered a situation where the doubly curved lens box is rigid-foldable individually, but the assembly of different lens boxes is no longer rigid-foldable?

Tendon Routing: Are the number and orientation of tendons optimized in the experiments, or are they distributed based on engineering intuition?

Fig. 4C shows that as the pre-tension linearly increases, the flexural modulus grows exponentially. This observation is quite surprising. Can the author elaborate on why? What is the physical underpinning of this behavior? Is there a threshold on tendon pre-tension, beyond which the flexural modulus can no longer increase?

I am not sure if the big-picture comparison in Fig. 5a is entirely fair, as other published studies did not utilize tendon-locking to increase their modulus. Perhaps the authors could display two curves based on their new folding pattern, one with the tendon and one without.

Version 1:

Reviewer comments:

Reviewer #1

(Remarks to the Author)

The authors have addressed my comments, and I can recommend the manuscript for publication now.

Reviewer #2

(Remarks to the Author)

I believe the authors have effectively addressed my comments and clarified the ambiguities in the original manuscript. Therefore, I support its publication.

Open Access This Peer Review File is licensed under a Creative Commons Attribution 4.0 International License, which permits use, sharing, adaptation, distribution and reproduction in any medium or format, as long as you give appropriate credit to the original author(s) and the source, provide a link to the Creative Commons license, and indicate if changes were

made.

Response to Reviewers' Comments

Manuscript Number:	NCOMMS-25-57487
Manuscript Title:	Smooth, doubly curved origami shells with reprogrammable rigidity
Article Type:	Research Article (Physical Sciences/Engineering)
Authors:	Morad Mirzajanzadeh and Damiano Pasini

We thank the reviewers for their valuable feedback and constructive comments. Our detailed response (light blue) to each of their remarks is provided below along with our revised portions of the manuscript (dark blue and indented).

Reviewer #1:

This manuscript presents a new origami structure called the doubly curved lens-box, which combines curved and straight folds. By embedding pre-tensioned tendons, the system enables reversible transformations from soft, compliant states to rigid, load-bearing structures with tunable stiffness. The authors present an inverse design framework for generating tessellations that conform to prescribed curved surfaces and validate the concept with paperboard prototypes, showing potential for applications in fields such as exoskeletons, deployable aerospace structures, robotics, and adaptive architectural systems.

The idea is interesting and the presentation is generally clear. I have some suggestions for the authors to improve the manuscript:

Comment # 1-1:

It is strongly suggested that the authors stick to the terminology of “rigid-ruling foldability,” rather than using rigid foldability and rigid-ruling foldability interchangeably, because by the most well-accepted definition, curved origami cannot be rigid foldable because it necessarily involves panel bending.

Response: We thank the reviewer for this suggestion. We have revised the manuscript to use ‘rigid-ruling foldability’ consistently, replacing ‘rigid foldability’ where appropriate. We retain ‘rigid foldability’ only when referring to straight-crease cases, such as the connector units, or quoting prior literature.

Comment # 1-2:

Line 151-152: this claim is wrong: “This property has significant implications as it enables the use of even non-deformable materials to create rigid-foldable origami structures...” Again, curved origami necessarily involves panel bending – no matter the amount, which is impossible for strictly non-deformable materials to achieve. Therefore, a softer tone is needed, such as saying “even stiff materials.” This is a problem throughout the manuscript. The authors have to be very careful of making strong claims as the proposed designs are not strictly “rigid origami.”

Response: Thank you for pointing out this seemingly contradictory statement.

While a curved-crease origami with non-deformable materials/panels is not expected to bend and fold, our approach circumvents this global rigidity by discretizing the curved creases into ruling-line segments, effectively converting the system into its equivalent, straight-crease origami composed of rigid panels. This approximation preserves the overall folded geometry and kinematics. By introducing torsional stiffness at the hinge locations corresponding to the original ruling lines, we can also emulate the bending response of a curved-crease origami that is made of thin, deformable panels. As a result, the shape and mechanical behavior of the original deformable-panel origami can be reproduced (approximately) in its thick-panel/non-deformable-panel, rigid-foldable counterpart. To clarify this, we have revised the statement as follows:

“This property has significant implications as it enables the use of even non-deformable materials to create the discretized, ruled segmentation version of the rigid-ruling foldable origami transitioning from a flat state to a locked, approximately curved shape.”

Please note that we also used similar clarifications in our original discussion section, as it reads:

“Demonstrated as rigid-ruling foldable, the discretized, ruled segmentation version of our patterns can be adapted to non-deformable materials or thick-panel origami.”

Additionally, we have checked the entire manuscript to make sure the claims/statements that have been made are clear and compatible with curved-crease origami restrictions, as recommended.

Comment # 1-3:

The authors mentioned about difficulties during folding, and indicated that some geometric parameters lead to deviation from rigid-ruling foldability. It is then important to investigate such phenomenon by conducting FEA, or bar-and-hinge model analysis, at least on a few particular patterns as preliminary results. It is not ok just saying such critical study is out-of-scope, when this matters to the claims in this manuscript.

Response: We thank the reviewer for highlighting this important point. We agree that this particular aspect deserves additional analysis and explanation. Although we briefly discussed this in the original SI, we now clarify the underlying mechanism yielding complex folding at certain angles in

the revised version of the manuscript and SI. We first explain the challenges associated with FE simulations attempting to model the folding of our double-curvature lens-box tessellation, and why current FE/bar-and-hinge modelling approaches are not yet well suited to systematically investigate rigid-ruling foldability and the folding complexity we observe for certain angles.

Rigid-ruling foldability and limitations of FE/bar-and-hinge models.

Rigid-ruling foldability is a *geometric property*: the ruling pattern must remain invariant throughout the motion. Standard FEA or compliant bar-and-hinge models, by design, allow panel stretching and non-guided bending, i.e., bending in directions other than ruling; these deformations enable motions that violate ruling invariance and isometric deformation. Hence their sole use cannot serve to test rigid-ruling foldability. For that reason, our endorsement of rigid-ruling foldability relies on geometric analysis and kinematic simulation tools that enforce isometric deformations and prescribed rulings (Freeform Origami). The challenge in assessing the rigid-foldability of multi-vertex origami patterns, including ours, has also been reported in another work (60) as mentioned in the original manuscript “Assessing the rigid-ruling foldability of a multi-vertex pattern such as our doubly curved lens-box tessellation is NP-hard (60).”

Folding analysis struggles for FE analysis.

Our building blocks comprise lens units and waterbomb-family connector units. The kinematics of a single lens unit is relatively straightforward *only if* it is idealized as a 1-DOF mechanism and if we impose our assumptions including folding symmetry and rigid-ruling foldability, as discussed in the manuscript and SI. In folding a physical prototype, however, it is often possible to accidentally bend panels along alternative ruling patterns, thus it is challenging to maintain rigid-ruling foldability and folding symmetry. The existence of several admissible ruling patterns implies that a lens unit effectively has a manifold of DOFs and admissible kinematic paths.

The connector units have an even more complex (3-DOF) rigid-folding motion, which we illustrate with the configurational space plotted in Supplementary Fig. 9b (shown again below) for a simplified, symmetric-folding case (reduced to 2 DOFs). The surface plot reveals the unbounded nature of the kinematic paths: any line (slice) on this surface corresponds to a possible (symmetric) folding path. For non-symmetric folding, which often occurs during physical paper folding, this design space becomes even richer. The blue line shown in Supplementary Fig. 9b represents only one of these possible paths, where a linear relation between certain folding angles exists (up to the turning point marked by the open circle in Supplementary Fig. 9d, left panel). In tessellated origami patterns, the number of DOFs and the range of admissible kinematic paths increase substantially, making the analysis and characterization of the folding process considerably more complex.

Fig. S9. Full-range configuration space of 2-DOF waterbomb connector unit.

Given the complex kinematic landscape described above, it is extremely challenging to prescribe boundary conditions in the FE model of our initially flat multi-DOF origami pattern so that all crease rotations evolve along one specific theoretical path of folding. The situation is even more complex in FE models with deformable thin materials, where rigid-ruling foldability can be easily violated, and several additional folding paths become accessible during folding. Relaxing any of the idealized assumptions leads to the emergence of kinematic paths that are unknown and therefore not directly comparable with our analytical model. In contrast, for single-DOF origami patterns with a

unique folding path, it is relatively straightforward to simulate the folding process, as is often demonstrated in the literature.

In our view, the main challenge for FE or bar-and-hinge simulations is to address an inverse problem: determining the boundary conditions and actuation (location, magnitude, and sequence) that will drive the system along a targeted folding path in a high-dimensional configuration space. To our knowledge, this inverse problem has not yet been systematically addressed in the literature and constitutes a challenging research direction in its own, with potential impact on revealing paths for automated folding of complex origami patterns. Existing rigid-folding simulation tools, such as Freeform Origami (also used in this work), rely on a geometric/kinematic constraint solver (but not a physical force-based folding simulator). They cannot provide direct information for analyzing the specific folding mechanics or complexities we observe.

While we could, in principle, force FE simulations to follow a theoretically possible kinematic path by prescribing specific rotational boundary conditions for individual connectors (hinges) with non-monotonic amplitude and thereby bring the origami into its folded state, such an analysis would artificially bypass the practical folding problem and thus obscure the *folding difficulty* raised by the reviewer. Conversely, if we prescribe the rotation of only a few creases and leave all other creases to fold freely, it becomes practically impossible for the FE model to find the experimentally observed (or desired) folding path in this high-dimensional configuration space.

To address the reviewer's concern within these limitations, rather than forcing an FE model along an imposed path, we now provide a detailed kinematic explanation of the observed folding difficulty, supported by additional parametric exploration of the configuration space (Supplementary Fig. 9d and related discussion).

Underlying mechanics of the folding difficulty.

Figure S9D illustrates the evolution of the folding angles $-\rho_1$, $-\rho_2$, and ρ_5 as functions of ρ_3 for four different sector angles β . For $\beta \geq \pi/2$, the folding angles $-\rho_1$ and ρ_5 increase monotonically with ρ_3 . In contrast, for $\beta < \pi/2$, the angles $-\rho_1$ and ρ_5 initially increase monotonically up to a turning point (marked by the open circle, corresponding to the contact point between the tail panels) and then decrease immediately, before rising again. In physical terms, the angles first rotate in one direction, reach a maximum or minimum at the turning point, and then continue rotating in the opposite direction. In other words, $-\rho_1$ cannot be treated as a smooth, globally monotone function of ρ_3 at that configuration. This indicates a kinematic singularity in the configuration space when the motion is parameterized by ρ_3 , even though the underlying rigid-folding motion remains continuous in the configuration-parameter space.

The above qualitatively explains why folding becomes more difficult than in the monotonic cases ($\beta = 90^\circ$ or 110°). More specifically, the following are the reasons:

- If we try to “drive” the mechanism mainly by actuating a panel that primarily changes ρ_3 , then near the turning point, crease 1 and crease 5 tend to halt and then reverse their rotation direction. If the manual actuation does not accommodate this reversal cleanly, the panels tend to bend, or the origami appears to “lock”.
- For monotonic curves, each value of ρ_3 corresponds to a unique configuration. For curves with a turning point, some configurations with the same $-\rho_1$ and ρ_5 correspond to distinct values of ρ_3 , reflecting the more complex, multi-valued relation between the folding angles.

- The turning point is associated with a singular posture when the motion is parameterized by ρ_3 . The mechanism remains kinematically admissible, but small imperfections in motion or loading can deviate it from the ideal rigid-folding path, inducing additional panel bending, and render the folding somewhat “fussy”.
- In paperboard origami with finite thickness and crease stiffness, this turning point can correspond to an energy peak or plateau. When manually folded, the origami may be perceived as undergoing a “stall” or a subtle snap-through event as the mechanism traverses the singular configuration

We agree that a dedicated FE/bar-and-hinge study, aimed specifically at solving the inverse actuation problem for such multi-DOF patterns, would be a valuable future work, but it lies beyond the scope of the present study.

To address the reviewer’s concern, we have expanded the SI (added the new **Section 3.3.2. ‘Kinematic Origin of Folding Difficulty’**) and now provide a detailed analytical explanation of the folding difficulty we observed.

“3.3.2 Kinematic Origin of Folding Difficulty

The two sets of Eqs. (33) and (36) trace a feasible 1-DOF rigid-folding motion of our connector waterbomb unit after the initial panel contact (red dot), corresponding to the second branch in Fig. S9b and to the curve beyond the red dot (and unfilled circles) in Fig. S9d. Our results show that the sector angles $\beta > \pi/2$ yield a strictly monotonic (increasing or decreasing) evolution of all dihedral angles along this path. However, for $\beta < \pi/2$, the dihedral angles ρ_1 and ρ_5 are no longer monotonic: they first change in one direction, then pass through an extremum and reverse.

During the manual folding process, we observed that while a single unit can be folded relatively easily for almost any β , folding a tessellated pattern becomes significantly more difficult for smaller sector angles $\beta < \pi/2$. We associate this increased difficulty with the kinematics of folding an origami pattern, where some creases exhibit a non-monotonic evolution of their dihedral angles, i.e., the folding rate $d\rho_i/d\rho_3$ (with $i = 1, 2, 5$) changes its sign along the motion (see Fig. S9d). At these turning points, ρ_i can no longer be treated as a smooth, globally monotonic function of ρ_3 ; the configuration corresponding to the extremum is a kinematic singularity when the motion is parameterized by ρ_3 . In practice, this means that as ρ_3 continues to change, certain creases must momentarily “stall” and then reverse rotation, which is difficult to realize through simple manual actuation and tends to promote panel bending or apparent locking.

In a tessellated lens-box pattern, many connector units share panels and must fold compatibly. For $\beta < \pi/2$, every connector must pass through its non-monotonic turning point, meaning that several creases across the tessellation must stall and reverse rotation in a coordinated way. Small mismatches between neighboring units then accumulate, forcing additional panel bending and local deviations from the ideal rigid-folding path, which manifest as apparent locking or “fussiness” during manual folding. We now turn our attention to assessing the rigid-foldability of a full lens-box pattern.”

We have also revised the related part of the manuscript as

“In particular, we use *Freeform (61)* rigid origami simulations to demonstrate that our pattern deforms **isometrically** and remains rigid-ruling foldable once tessellated in plane (see Supplementary Note 3.5). To achieve this, we first approximate the curved-crease fold pattern with a piecewise linear version consisting of a finite number of ruling segments, forming a *discrete ruled surface*. While the mobility of the resulting pattern remains independent of the ruling discretization, its folded geometry converges to the smoothly folded curved-crease shape as the number of ruling segments increases. Although we cannot **fully** analyze the folding **mechanics** of the tessellated pattern through rigid-folding simulations **alone**, we observe that in practice, folding becomes **more challenging when the sector angle $\beta < \pi/2$** (Fig. 1a-top), and the folding difficulty increases as β decreases. We attribute this phenomenon to the kinematics of the origami pattern, where some creases exhibit a non-monotonic evolution of their dihedral angles and sharply reverse their folding direction. While this behavior is manageable for a single unit, it becomes strongly amplified when several connector units are tessellated and required to fold compatibly (see Supplementary Note 3.3.2 and Supplementary Fig. 9).”

Comment # 1-4:

On a related note, in the constrained optimization, the authors should report how much the constraints are violated in the end, which is very likely, as in Ref. 20.

Response: As suggested by the reviewer, we have now quantified the residual constraint violation and stationarity quality for all optimized lens-box units.

For each of the optimized lens-box geometries, we evaluated every constraint at the final solution p^{i*} . For nonlinear equality constraints $\Phi^j(p^i) = 0$ (Eq. (3), Eq. (20), and Eq. (43)), the maximum absolute residual $|\Phi^j(p^{i*})|$ is always below 10^{-12} . All nonlinear inequality constraints $\Psi^k(p^i) \leq 0$ (e.g., geometric bounds, angle limits, non-penetration, no-overlap, and rigid-ruling foldability) and all simple variable box bounds ($p_a^q \leq p_q^i \leq p_b^q$) are satisfied with zero violation within numerical precision.

We also assessed first-order optimality. For each optimized design, we restarted the interior-point solver from that solution and recorded the solver diagnostics. The solver-reported feasibility residuals are on the order of 10^{-15} – 10^{-16} , and the KKT (Karush–Kuhn–Tucker) (stationarity) residuals are on the order of 10^{-8} . The complementarity residuals are on the order of 10^{-20} or smaller.

These results indicate that our final solutions satisfy all constraints up to numerical precision and meet the KKT optimality conditions within $\mathcal{O}(10^{-8})$.

To address the reviewer’s comment, we have added this quantitative assessment to the SI:

“We implement the numerical optimization algorithm in Mathematica using the interior-point **method (FindMinimum)**. The initial guesses for the design variables are **chosen based on our understanding of the problem and the geometric attributes of the lens box unit cell**; they are then refined iteratively to approach a feasible solution.

The optimization is considered successful if the algorithm returns a solution that satisfies all constraints within a given numerical tolerance (i.e., the maximum residual among all constraints is smaller than 10^{-8}) over several hundred iterations.

After convergence, we evaluate the numerical quality of each solution in two ways:

(i) *Feasibility*. We substitute the optimized design variables p^{i*} back into all constraints and measure any violation. For each equality constraint $\Phi^j(p^i) = 0$, we compute the residual $|\Phi^j(p^{i*})|$ across all unit-cell designs. The maximum equality residual is below 10^{-12} . All inequality constraints $\Psi^k(p^i) \leq 0$ (e.g., geometric bounds, angle limits, non-penetration, no-overlap, and rigid-ruling foldability) and all simple variable box bounds ($p_a^q \leq p_a^i \leq p_b^q$) are satisfied with zero violation within numerical precision.

(ii) *First-order optimality*. We restart the interior-point solver from the optimized solution and record the solver diagnostics. The solver-reported feasibility residuals are on the order of $10^{-15} - 10^{-16}$, the KKT (Karush-Kuhn-Tucker) (stationarity) residuals are on the order of 10^{-8} , and the complementarity residuals are on the order of 10^{-20} or smaller.

These values indicate that our designs are numerically feasible to machine precision and satisfy the KKT optimality conditions at the $\mathcal{O}(10^{-8})$ level.”

Notation update. We now write inequality constraints in the standard ≤ 0 form (equivalent to the previous > 0 convention via negation). This change is purely notational and does not affect the feasible set of results.

Comment # 1-5:

Line 246-247: It should be clearly indicated in Fig. 2 about whether a pattern is folded from an entire piece or glued from many separate parts. The cut-and-glue strategy is fine, but it is important to clearly show this to the readers.

Response: We thank the reviewer for the helpful suggestion. We have revised the manuscript and Fig. 2 caption to explicitly state the fabrication method for each prototype. The revised manuscript now includes:

“Figure 2a-g shows the results of our analysis applied to a geometrically diverse set of cylindrical and doubly-curved surfaces, with paperboard prototypes fabricated through laser perforation and manual folding. Cylindrical surfaces are folded from a single sheet (no adhesive), whereas doubly curved surfaces are assembled by folding strips into layers in the x - y plane and bonding along the z -direction.”

The caption of Fig. 2 now includes the following statement:

“In panels (a–c), the origami patterns are folded from a single sheet. In panels (d–g), the patterns are assembled by folding strips into layers in the x - y plane, which are then glued along the z -direction to form the complete surface.”

Comment # 1-6:

All demonstrations in this work are made of paperboards. However, paperboards are quite forgiving for stretching deformation, which, by theory, should not present in this work. Therefore, it is important to show material versatility of the proposed designs by trying a different material, especially materials with large in-plane stiffness.

Response: We appreciate this suggestion. To further demonstrate the potential of our origami fabricated with other materials, we would like to share unpublished prototypes that successfully demonstrate doubly curved lens-box folding using materials with high in-plane stiffness (to minimize the possibility of in-plane stretching) in the pictures below. These prototypes are fabricated from an aramid–epoxy laminate $[(0/90)_s]$ which has a typical in-plane tensile modulus of about 40 – 70 GPa (around an order of magnitude larger than paperboard), ensuring that the kinematics are not reliant on material stretch. The results show that it is feasible to fold our doubly curved lens-box patterns from these composite sheets by only adjusting the laser-cutting parameters to create creases that fold without failure. While these efforts are ongoing, we share these early results with the reviewer to demonstrate the feasibility of folding our patterns using materials with very high in-plane stiffness, thus better matching the theoretical assumptions used in our paper.

[Figure Redacted]

Comment # 1-7:

There are very little explanations for Fig. 4f-h. Please elaborate on how the different configurations are stabilized.

Response: Thank you for the helpful comment. In these demonstrations, the tendon–origami structure remains compliant because tendons are not fully pre-tensioned.

- Fig. 4f and g: The origami shapes are stabilized by panel bending and unit self-contact under a light boundary constraint at the edges (manual holding during the demo).
- Fig. 4h: The configuration is free-standing, arising from a balance of gravity, partial (non-uniform) tendon pre-tension, panel bending rigidity, and unit contact. Fully pre-tensioning the tendons leads to the final locked state shown as configuration ⑦. Each intermediate increment (not measured quantitatively) corresponds to a specific distribution of pre-tension in the top and bottom tendons.

To make this clear, we have added the following statement in the revised manuscript:

“In Fig. 4f and g, the tendon-origami shell is stabilized by panel bending and unit self-contact under a moderate boundary constraint at the edges (manually held during the demonstration). In Fig. 4h, the shell attains free-standing equilibrium states (① - ⑦) where gravity is balanced by partial, non-uniform tendon pre-tension, panel bending rigidity, and unit contact; a full pre-tension of the tendons produces the final locked configuration (⑦).”

Reviewer #2:

This paper presents a new origami folding pattern -- called "doubly curved lens-box" that can develop a flat sheet precursor into a doubly curved surface (with a smooth curvature and piecewise curvature). This is a significant step forward from the author's previous paper, which showed only a single curved surface. In addition, the authors proposed a novel method of locking the folded curvature with embedded tendons, demonstrating the practical use of this new origami pattern for different engineering functions.

Overall, this paper presents an impressively thorough framework for the design and kinematic analysis of the doubly curved lens-box. The tendon locking is a nice addition. Therefore, I recommend publication with the following suggestions for further improvement.

Comment # 2-1:

Reachable design space: Although the authors have demonstrated several fascinating examples in Fig. 2, it is never entirely clear what range of curvatures the doubly curved lens-box can achieve. Is there a theoretical limit on the obtainable κ_I and κ_{II} ?

Response: Thank you for the thoughtful question about the attainable values of curvature. Our origami approach is scalable: in the zero-thickness, inextensible-panel idealization, arbitrarily large curvatures (small radii) are achievable by refining the crease pattern. In practice, though, the realizable curvature is limited by the thin-panel assumption we made for the base sheet and by the effective global thickness of the folded origami shell.

To remain in the bending-dominated regime and avoid in-plane stretch, we require (i) a small **sheet-thickness-to-crease-length** ratio, e.g., $t/\ell_{\text{crease}} \lesssim 0.01 - 0.05$ (thin-plate range), and (ii) a small thickness-to-radius (of the bent panel) ratio, e.g., $t/R_c \lesssim 0.01$ (equivalently, surface strain $\varepsilon \approx t/(2R_c) \ll 1$) [R1]. As curvature increases (i.e., R_c decreases), these constraints typically necessitate either finer patterns and/or thinner base sheets. Therefore, both the base-material thickness t and the global thickness of the origami shell determine the practically attainable envelope of curvature values.

If the reviewers' question seeks a numerical curvature range for a given sheet thickness, that is material- and pattern-dependent, this lies outside the scope of this paper; a general closed-form bound is non-trivial and remains an open question of research.

To address the reviewer's comment, we have now included the following statement in the manuscript:

“Scaling the origami pattern enables to determine its curvature design space; however, in practice, the attainable curvature is bounded by thin-panel criteria on the base sheet (e.g., $t/\ell_{\text{crease}} \lesssim 0.01 - 0.05$, $t/R_c \lesssim 0.01$, where ℓ_{crease} is a characteristic crease length). Achieving larger curvature (smaller R_c) generally requires finer tessellations and thinner base material, which in turn reduces the global thickness of the folded shell.”

[R1] Calladine, C.R., 1983. *Theory of shell structures*. Cambridge university press.

Comment # 2-2:

Rigid foldability: Is local rigid foldability only a necessary condition for global rigid foldability? That is, have you encountered a situation where the doubly curved lens box is rigid-foldable individually, but the assembly of different lens boxes is no longer rigid-foldable?

Response: Yes. Unit-level (local) rigid-(ruling) foldability is a **necessary but not sufficient** condition for the global rigid foldability of a tessellation. We have encountered configurations where an individual doubly curved lens-box is rigid-(ruling) foldable, yet specific 2D assemblies are not, due to kinematic incompatibilities at shared creases (e.g., closure constraints imposed by periodicity).

That said, the outcome depends on the assembly. When lens-box units are tessellated only along their longitudinal (y) direction (Fig. 1a), the inter-unit kinematics matches, and the unit-level rigid-(ruling) foldability is sufficient to obtain global rigid-(ruling) foldability. More complex 2D tessellations can introduce additional compatibility constraints that may break global rigid foldability.

As noted in the main manuscript, in general, “Assessing the rigid-ruling foldability of a multi-vertex pattern such as our doubly curved lens-box tessellation is NP-hard (60).”, which explains why sufficiency can fail in some assemblies.

Comment # 2-3:

Tendon Routing: Are the number and orientation of tendons optimized in the experiments, or are they distributed based on engineering intuition?

Response: We selected the tendon count and routing using a simple, rigidity-driven design procedure rather than ad-hoc intuition. Specifically, we (i) ensured structural rigidity of the locked lens-box unit under compression, and (ii) minimized the number of tendons subject to manufacturability constraints (placing holes as close as the practical limit to the origami vertices and along fold-compatible directions). This yields a sparse, symmetric routing that stabilizes the unit while limiting non-rigid deformations in the lock state.

Here, we did not perform a global combinatorial optimization over all possible tendon counts and locations. This means other routings could also work and may yield different rigidity/stiffening trade-offs; systematically exploring these is an interesting direction for future work.

Comment # 2-4:

Fig. 4c shows that as the pre-tension linearly increases, the flexural modulus grows exponentially. This observation is quite surprising. Can the author elaborate on why? What is the physical underpinning of this behavior? Is there a threshold on tendon pre-tension, beyond which the flexural modulus can no longer increase?

Response: Thank you for this helpful question.

Mechanism. The tested panel (pre-stressed shell) contains two tendon sets (top and bottom). In our experiment, the bottom tendons were incrementally stretched at every step (nominally linear), while the top tendons were additionally stretched only at selected steps, i.e., 4,7,9 and 10, to re-flatten the unloaded shell (as noted in the original Fig. 4b-c caption). This staggered engagement raises the overall pre-stress, and increases the apparent bending stiffness via in-plane-bending coupling (stress stiffening). Because the top-tendon stretches occur in discrete events (and with larger magnitudes at later steps), the effective pre-stress (top + bottom) grows convexly with the step count rather than strictly linearly. The apparent flexural modulus, therefore, follows a convex trend that is well fit by an exponential curve.

Threshold. We halted pre-tensioning at the final reported point because further stretching produced visible cell distortion, local panel buckling, and large out-of-plane bending; beyond this onset of instability, additional pre-tension did not yield any further reliable increase in the apparent flexural modulus for this configuration. The precise threshold depends on pattern geometry, tendon routing, and material properties, and a rigorous value would require a dedicated stability analysis (e.g., tracking the first occurrence of local wrinkling/buckling or a slope change in the stiffness-pre-tension curve).

To clarify these points, we have revised the related part in the manuscript into:

“Our results in Fig. 4c show that the apparent flexural modulus increases convexly with the applied tendon pre-tension due to the (i) in-plane-bending coupling (geometric/stress stiffening) in the pre-stressed shell and (ii) staggered engagement of the top tendons at steps 4, 7, 9, and 10 to re-flatten the unloaded shape. This nonuniform engagement causes the effective pre-stress (top and bottom) to rise nonlinearly with the nominal increment in the bottom tendon pre-tension, yielding an exponential-like trend in the bending stiffness. We terminated the loading at the final reported point, where additional pre-tension triggered cell distortion and local buckling. Beyond this onset of instability, further pre-tension did not translate into a reliable stiffness increase for this configuration. (The lowest modulus value remained unregistered due to the challenge of measuring the initial tendon length in a formless relaxed state.)”

Comment # 2-5:

I am not sure if the big-picture comparison in Fig. 5a is entirely fair, as other published studies did not utilize tendon-locking to increase their modulus. Perhaps the authors could display two curves based on their new folding pattern, one with the tendon and one without.

Response: Thank you for noting this. Our original plot demonstrates both trends, i.e., the origami alone without tendon stretch (**dashed** curve) and the tendon-stiffened case (**solid** curve). This is why our pattern's stiffness curve starts from a small bending rigidity; we also show that it can be continuously increased by applying tendon stretching, though this progression is not explicitly depicted on the original plot. In the initial stages, our stiffness remains at its lowest value, even among the literature, due to our floppy tendon-origami structure with multiple DOFs.

We also emphasize that the literature shown here is not readily compatible with integrating a similar tendon mechanism: A necessary condition for a comparable approach is the presence of contact-based locking mechanisms within the origami structure.

To address the reviewer's comment, we have clarified the original data point representing our work **without tendons** ($T=0$; large light blue circle) and highlighted the application of tendons and its **increasing pre-tension** ($T > 0$) effect on stiffness in our revised Fig. 5a, as shown below:

We also revised the first sentence of the Discussion to highlight that prior literature does not use tendons:

“This work has introduced a hybrid origami tessellation combining straight and curved creases, capable of isometric folding and locking into structural shells with smooth doubly-curved or cylindrical surfaces with varying curvature, thereby addressing the trade-off between flexural rigidity and curvature smoothness in traditional tendon-free origami tessellations (see Fig. 5a).”

The caption of Fig. 5 has been revised to:

“Fig. 5. Trade-off between rigidity and curvature smoothness, and potential applications. a Qualitative rigidity-smoothness plot for surfaces obtained from existing tendon-free origami tessellations, illustrating...”